# Precision Surgery in NSCLC

**DOI:** 10.3390/cancers15051571

**Published:** 2023-03-03

**Authors:** Giorgio Cannone, Giovanni Maria Comacchio, Giulia Pasello, Eleonora Faccioli, Marco Schiavon, Andrea Dell’Amore, Marco Mammana, Federico Rea

**Affiliations:** 1Thoracic Surgery Unit, Department of Cardiac, Thoracic Vascular Sciences and Public Health, University Hospital of Padova, 35128 Padova, Italy; 2Medical Oncology 2, Veneto Institute of Oncology IOV IRCCS, 35128 Padova, Italy; 3Department of Surgery, Oncology and Gastroenterology, University of Padova, 35128 Padova, Italy

**Keywords:** NSCLC, immunotherapy, targeted therapy, precision surgery

## Abstract

**Simple Summary:**

The introduction of new therapies for non-small cell lung cancer (NSCLC) has radically changed the point of view of thoracic surgeons, leading them to pay increasingly more attention not only to the clinical stage, but also to the genomic and molecular features of the disease and the potential for multimodality treatments. This is the concept of precision surgery in thoracic oncology. The aim of our paper is to summarize the changes in thoracic surgical practice that occurred after the introduction of immunotherapy and targeted therapy for the treatment of NSCLC.

**Abstract:**

Non-small cell lung cancer (NSCLC) is still one of the leading causes of death worldwide. This is mostly because the majority of lung cancers are discovered in advanced stages. In the era of conventional chemotherapy, the prognosis of advanced NSCLC was grim. Important results have been reported in thoracic oncology since the discovery of new molecular alterations and of the role of the immune system. The advent of new therapies has radically changed the approach to lung cancer for a subset of patients with advanced NSCLC, and the concept of incurable disease is still changing. In this setting, surgery seems to have developed a role of rescue therapy for some patients. In precision surgery, the decision to perform surgical procedures is tailored to the individual patient; taking into consideration not only clinical stage, but also clinical and molecular features. Multimodality treatments incorporating surgery, immune checkpoint inhibitors, or targeted agents are feasible in high volume centers with good results in terms of pathologic response and patient morbidity. Thanks to a better understanding of tumor biology, precision thoracic surgery will facilitate optimal and individualized patient selection and treatment, with the goal of improving the outcomes of patients affected by NSCLC.

## 1. Introduction

Lung cancer still represents one of the most common causes of cancer related death worldwide despite significant advances in research [1]. About 80–85% of lung cancers are represented by non-small cell lung cancer (NSCLC) [2]. NSCLC is a heterogeneous disease and the two main histologic subtypes are represented by adenocarcinoma and squamous cell carcinoma. However, many other NSCLC subtypes exist (e.g., pleomorphic carcinoma, mucoepidermoid carcinoma, sarcomatoid carcinoma). Indeed, the most recent WHO classifications are gaining increasing complexity, and have now included immunohistochemistry and molecular testing, together with morphological analysis, in the NSCLC definition. This has allowed a more precise differentiation of the histologic subtypes of lung cancer, thus leading to improved therapeutic strategies [3].

Surgery is the mainstay of therapy for early-stage NSCLC, leading to cure in the majority of patients. However, despite the recent introduction of lung cancer screening, the majority of tumors are diagnosed at advanced stages, where treatment options are limited. In the era of conventional cytotoxic drugs, the prognosis for this cancer was grim. The discovery of the central role of the immune system and the development of new molecular alterations have radically changed the approach to lung cancers. Recent research in the tumor biology of lung adenocarcinoma and squamous cell carcinoma has shown significant differences in the tumor immune microenvironment [4]. Moreover, in recent years, since the discovery of tumor cells’ ability to release macromolecules into the bloodstream, the role of liquid biopsy has gained increasing importance. Indeed, in selected cases, liquid biopsy has replaced tissue sampling when suspecting the onset of resistance to new targeted drugs [5].

With the introduction of new drugs, such as immune checkpoint inhibitors (ICIs) and targeted therapies, the prognosis of some advanced NSCLCs have changed.

Moreover, a new age of precision oncology medicine has begun as a result of the development of new diagnostic technology and bioinformatics tools that have added to our understanding of cancer biology.

This is based on the idea that each tumor is unique and can be precisely targeted by one of the many targeted drugs reportedly available [6]. In fact, the recommendation of particular therapeutic modalities considers the genetic and environmental influences on therapeutic response [7]. The term “precision medicine” was introduced because diagnostic, prognostic, and therapeutic approaches are meticulously tailored to each patient’s needs.

These novelties have radically changed the world of surgical oncology and, similar to precision medicine, the new concept of precision surgery is proposed.

It was firstly introduced in general surgery, especially related to the different prognoses of colorectal cancer related to different genetic mutations. Nowadays, this concept is growing in thoracic oncology, too, especially with the discovery of the crucial role of molecular alterations in NSCLC. We herein describe the most important molecular alterations and the different clinical trials of targeted therapies for NSCLC, highlighting their implications in thoracic surgery, and we introduce the new concept of precision surgery in lung cancer.

## 2. Molecular Alterations and TKIs Trials

### 2.1. Epidermal Growth Factor Receptor-TKIs Trials

Nowadays, the ideal management of NSCLC tumors requires the analysis of a series of biomarkers that may aid in determining target therapy sensitivity. Most research on lung adenocarcinoma during the past ten years has concentrated on mutations of the epidermal growth factor receptor (EGFR). EGFR is a member of a family of receptor tyrosine kinases that can set off several signaling cascades that promote cell growth and proliferation. About 20% of patients with lung adenocarcinoma have EGFR mutations, such as exon 19 deletions and exon 21 (L858R) point mutations. Gefitinib and erlotinib are reversible competitive inhibitors of ATP for the tyrosine kinase domain of EGFR and they represent the first generation of EGFR TKIs. On the contrary, the second generation drug afatinib irreversibly inhibits human epidermal growth factor receptor (EGFR) kinases and has more targets than the first generation [8].

Different clinical trials have demonstrated the great advantages of these molecules over standard chemotherapy for advanced NSCLC. 

In the OPTIMAL, CTONG-0802 trial, patients with histologically confirmed stage IIIB or stage IV NSCLC (according to the 6th edition of TNM) and an activating mutation of EGFR (exon 19 deletion or exon 21 L858R point mutation) were randomly assigned to receive oral erlotinib or up to four cycles of gemcitabine plus carboplatin. Erlotinib patients had considerably longer median progression-free survival (PFS) than chemotherapy patients (13.1 vs. 4.6 months), and chemotherapy was linked to more grade 3 or 4 adverse effects than erlotinib [9].

According to the trials LUX-lung 3 and LUX-Lung 6, afatinib outperformed traditional chemotherapy in stage IIIB or IV (7th edition of TNM) lung cancer adenocarcinoma with del19 EGFR mutation (pemetrexed-cisplatin in LUX-Lung 3 and gemcitabine-cisplatin in LUX-Lung 6, respectively). However, despite much longer PFS in LUX-Lung 3 and LUX-Lung 6, there was no difference in overall survival (OS) [10]. 

After these studies, both first- and second-generation TKIs were approved for first-line treatment of EGFR mutation-positive advanced non-small-cell lung cancer. 

Subsequently, patients with stage IIIB or stage IV NSCLC and a common EGFR mutation were enrolled in LUX-Lung 7, which compared the effectiveness and safety of afatinib and gefitinib (in exon 19 deletion or Leu858Arg). The results of this trial showed that afatinib significantly improved outcomes with a manageable tolerability profile in patients with EGFR-mutated NSCLC who had not previously received treatment as compared to gefitinib [11].

Unfortunately, after 10 years of treatment with first- and second-generation EGFR-TKIs, drug resistance developed in the majority of patients, primarily as a result of the exon T790 M mutation (exon 20).

Subsequently, osimertinib, a third generation EGFR-TKI, was released. Osimertinib is an irreversible tyrosine kinase inhibitor that inhibits T790M mutations. The AURA series trial examines the effectiveness and security of osimertinib. The phase 3 AURA3 study demonstrated significantly enhanced efficacy of osimertinib in comparison to platinum pemetrexed regimen in NSCLC patients who had gained the T790M resistance mutation after treatment with first generation TKI. The results showed that osimertinib had a considerably greater objective response rate (ORR) than combination chemotherapy (71% vs. 31%; odds ratio for objective response, 5.39; 95% CI, 3.47 to 8.48, *p* < 0.001) and a longer PFS (10.1 months vs. 4.4 months; hazard ratio (HR) = 0.3, 0.23 to 0.41, *p* < 0.001). Notably, osimertinib has an intracranial ORR of 70% [12].

On the view of these results, osimertinib was approved as a second-line treatment for advanced lung cancer with the EGFR T790M mutation. 

In the FLAURA trial, standard first-generation EGFR-TKIs were compared to first-line treatment with osimertinib. Osimertinib was well tolerated and significantly increased PFS (18.9 vs. 10.2 months; HR 0.46). Therefore, osimertinib was approved by the FDA in 2018 as the first-line treatment for patients with metastatic NSCLC who have either an EGFR exon 19 deletion or an exon 21 L858R mutation [13]. 

All these studies evaluated the role of EGFR-TKIs as the definitive treatment in advanced EGFR-mutated NSCLC. 

The ADAURA trial evaluated the efficacy of osimertinib in an adjuvant context [14]. This study assessed the efficacy of osimertinib in comparison to standard chemotherapy in patients with resected stage IB-IIIA NSCLC (according to the eighth edition of TNM) and a verified EGFR-activating mutation (exon 19 deletion or exon 21 L858R). Finally, 90% of patients with stage II-IIIA of the osimertinib group were still alive and disease-free at 2 years, as opposed to 44% of patients in the placebo group. The authors concluded that patients with stage IB-IIIA EGFR-mutated NSCLC who received osimertinib had a significantly better disease-free survival (DFS) compared to placebo.

### 2.2. ALK Translocation and ALK-TKIs Trials

The anaplastic lymphoma kinase (ALK) gene was first identified in 1994, when it was found to be fused to nucleophosmin in a subtype of non-Hodgkin lymphoma [15]. The identical ALK gene was discovered in NSCLC a few years later by Soda et al., and this time it was joined to Echinoderm Microtubule-Associated Protein-Like Protein 4 (EML4) [16].

About 3–7% of NSCLC patients have ALK gene rearrangements, frequently discovered without EGFR or KRAS mutations. The FDA has approved the use of three ALK tyrosine kinase inhibitors (TKIs), including crizotinib, ceritinib, alectinib, and brigatinib, for the treatment of NSCLC with an ALK rearrangement. Crizotinib was the first ALK inhibitor to be approved by the FDA. In the Profile 1014 study, crizotinib and chemotherapy were compared as first-line treatments for patients with advanced non-squamous NSCLC with ALK-gene mutation. With crizotinib, progression-free survival was considerably longer than with chemotherapy (median 10.9 months vs. 7.0 months).

Unfortunately, after a few months of treatment, acquired resistance to crizotinib will develop. Therefore, second-generation (ceritinib, alectinib, and brigatinib) and third-generation (lorlatinib) ALK inhibitors were introduced. In particular, alectinib has been demonstrated to have a high rate of intracranial efficacy in treating brain metastases. In the phase 3 randomized, open-label ALEX trial, 303 patients with advanced, untreated ALK-positive NSCLC were randomly assigned to receive either alectinib or crizotinib. Alectinib considerably outperformed crizotinib in terms of progression-free survival (12-month event-free survival rate; 68.4% vs. 48.7%). Central nervous system (CNS) progression was one of the secondary endpoints that was significantly lower in the alectinib group [17].

Brigatinib, a second-generation ALK inhibitor, outperformed crizotinib in the ALTA-1 study for PFS and health-related quality of life (QoL) in advanced ALK drug-naive ALK-positive non-small cell lung cancer. As a result of this study, brigatinib was approved as the initial therapy for individuals with advanced ALK-positive NSCLC [18]. 

First-line lorlatinib was evaluated against crizotinib in a phase 3 randomized trial named CROWN. An objective response was observed in 76% of patients in the lorlatinib group and 58% of those in the crizotinib group. However crizotinib was associated with fewer grade 3 or 4 adverse events [19].

### 2.3. EGFR/ALK-TKIs as Neoadjuvant Treatment for NSCLC

Following the great results of EGFR and ALK-TKIs in advanced NSCLC in terms of the increase in DFS and OS, different clinical trials are being conducted for these drugs in the neoadjuvant settings. Similarly to the adjuvant protocols, neoadjuvant TKIs may work by removing residual tumor cells or micrometastases created by primary tumor cells (with comparable genotypes and molecules), inducing greater clinical responses.

The surgical results of these trials are crucial because they could provide information into the viability and safety of lung resection following these new treatments. Neoadjuvant administration of EGFR and ALK-TKIs in early NSCLC currently has little experience. There are no published phase 3 randomized studies and the majority of the evidence comes from short case series. 

Neoadjuvant gefitinib was used in six patients with locally advanced lung adenocarcinoma, in a retrospective report by Zang et al. The authors determined that surgery was possible, but no information on pathological response, OS or PFS was provided [20].

Patients with stage IIIA-N2 disease were enrolled in the EMERGING-CTONG 1103 trial, where Zhong et al. compared targeted therapy to conventional neoadjuvant therapy (seventh edition of TNM) and an EGFR mutation. A total of 71 patients were recruited, of whom 37 were randomly assigned to receive erlotinib and 34 received cisplatin plus gemcitabine. Erlotinib-treated patients had a considerably prolonged PFS (21.5 months vs. 11.4 months in the chemotherapy group) [21]. 

The NeoADAURA trial (NCT04351555) is still ongoing. In this trial, neoadjuvant osimertinib as a monotherapy or in combination with chemotherapy is tested on patients with resectable EGFR-mutated NSCLC [22].

Ning showed that some patients with EGFR-positive advanced NSCLC were eligible for surgery following gefitinib therapy: progression-free survival was 14 months, and overall survival might reach 36 months [23]. 

In another trial (NCT01217619), the role of neoadjuvant erlotinib in patients with stage IIIA-N2 (according to the seventh edition of TNM) EGFR-mutated NSCLC was analyzed. The study enrolled a total of 19 patients, 14 of which underwent surgery. Pathological downstaging occurred in 21% of cases with a 68% incidence of radical resection [24]. 

ALK-positive NSCLC often has a worse clinical prognosis than EGFR-mutant NSCLC; in fact, it seems to be more aggressive and resistant to traditional antineoplastic drugs. 

Targeted therapy in the neoadjuvant and adjuvant settings is being studied in trials with resectable NSCLC that has an ALK mutation (ALCHEMIST, NCT02201992; ALNEO, NCT05015010).

Zhang et al. reported 11 patients with locally advanced NSCLC treated with neoadjuvant crizotinib. A full resection was possible since all 11 patients demonstrated a favorable response to induction therapy. Moreover, all patients showed good tolerance to neoadjuvant crizotinib [25].

For the moment, only case reports are available concerning the use of alectinib and the other second-generation ALK-TKIs in the neoadjuvant setting [26].

## 3. The PD-1/PD-L1 Pathway and Its Clinical Implications

Immunotherapy radically changed treatment options in the oncology world. The immune system works by maintaining a delicate balance between immunological checkpoint-mediated suppression (CTLA-4) and costimulatory mediators (CD28) of T cell activation. Programmed cell death 1 (PD-1) was discovered to be another inhibitory receptor, redefining the significance of immunological checkpoints in ensuring the preservation of T cell tolerance.

The discovery that PD-L1 overexpression on a mouse mastocytoma cell line inhibits CD8+ T cell cytolytic function by ligating PD-1, supported the theory that activation of the PD-1/PD-L1 pathway can decrease immune responses for malignancies, allowing increased tumor growth and invasiveness. Through antiapoptotic PD-L1-mediated signals, the PD-1/PD-L1 pathway promotes the survival of cancer cells in the tumor microenvironment [27].

Therapies targeting the immunological checkpoints programmed cell death-1 (PD-1), programmed cell death ligand-1 (PD-L1), and cytotoxic T-lymphocyte-associated antigen-4 (CTLA-4) have been approved for the treatment of a variety of tumor types, including NSCLC. Different drugs are now feasible, on the basis of the type of inhibitor activity. 

Although PD-L1 expression in tumors has been used to predict treatment response, the method’s sensitivity and specificity are very moderate. This is due to the fact that PD-L1 alone cannot fully reflect the heterogeneity of the tumor microenvironment that is involved in the response to immunotherapy. 

Patients with at least 50% PD-L1 expression, in the Keynote-024 research were randomized to either platinum-based chemotherapy or pembrolizumab, an anti-PD1 medication. In the pembrolizumab group, the median progression-free survival was 10.3 months compared to 6.0 months in the chemotherapy group. Moreover, compared to platinum-based chemotherapy, pembrolizumab was linked to a considerably longer progression-free and overall survival as well as fewer adverse events [28].

The extraordinary findings of the double-blind, placebo-controlled PACIFIC trial demonstrated that durvalumab (an anti-PD-L1) significantly improved PFS and OS in this heterogenous patient population while maintaining favorable safety profiles [29]. This trial represents the birth of the world of the new therapies applied in clinical practice.

In the OAK trial, atezolizumab and docetaxel were compared for effectiveness and safety in NSCLC patients with squamous and non-squamous cell histologies [28]. Patients were randomized to receive either atezolizumab or docetaxel every three weeks regardless of PD-L1 expression. In the atezolizumab arm, the median OS was longer (13.8 months vs. 9.6 months), and the effect persisted independently from PD-L1 expression. The patients with the highest levels of PD-L1 expression had the largest OS improvement (20.5 months vs. 8.9 months). Pembrolizumab, an anti-PD-1 drug, is currently authorized for use as first- and second-line therapy in patients with advanced NSCLC whose tumors exhibit PD-L1 expression according to immunohistochemical testing. Atezolizumab (anti-PD-L1) and nivolumab (anti-PD-1) are recommended as second-line therapies independently from PD-L1 expression. [30,31,32]. Patients with unresectable stage III NSCLC, whose disease has not progressed despite concomitant platinum-based chemoradiotherapy, are eligible to receive durvalumab (anti-PD-L1) as a maintenance therapy.

In the adjuvant setting, Impower010 is the first phase 3 study to demonstrate DFS improvement with adjuvant atezolizumab in completely resected stage IB to IIIA NSCLC (seventh edition of TNM) after platinum-based chemotherapy [33]. This trial was approved by EMA and showed significant DFS benefit in patients in stage II–IIIA whose tumors expressed PD-L1 on 1% or more. Particularly, those with PD-L1 > 50% appeared to gain the most benefit.

In patients with totally resected stage IB, stage II, and stage IIIA (seventh edition of TNM), the phase 3 PEARLS/KEYNOTE-091 trial showed DFS improvement with pembrolizumab compared with placebo, followed by adjuvant chemotherapy as recommended by guidelines [34]. 

### Neoadjuvant Immunotherapy or Chemo-Immunotherapy

Immune checkpoint inhibitors have been studied as a monotherapy or together with chemotherapy in the neoadjuvant setting. The existence of a macroscopic tumor can offer a greater variety of tumor neoantigens able to activate the immune system in the neoadjuvant scenario, encouraging the earlier elimination of micrometastases. Neoadjuvant chemotherapy results in an increase in PD-L1 positive tumor cells and immune infiltrates, which would support the potential synergy with immunotherapy. This concept justifies the use of combination therapy.

Chaft et al., firstly reported their surgical experience in patients with NSCLC previously treated with T cell checkpoint inhibitors [35]. A total of five patients deemed to be unresectable at diagnosis received ICIs treatment. In three patients, chest and PET-CT scans showed a local persistence of disease in the lung and mediastinal lymph nodes. For this reason, they were considered for surgery with debulking intent. The final pathological exam showed a significant response rate on the specimens, demonstrating a discrepancy between the radiologic and pathologic evaluation. The other two patients were oligometastatic because of an isolated adrenal gland and CNS metastasis, respectively. The CNS metastasis was firstly irradiated, then treated with ICIs and, finally, the patient underwent right lower lobectomy with major pathological response (pT1bN0). The other patient was firstly treated with ICIs, then surgically resected both on the adrenal gland (with a laparoscopy approach) and on the lung via a wedge resection (with robotic assisted VATS). The final pathological exams showed a complete pathological response (pT0N0M0) after treatment with ICIs.

Two years later, Bott et al. conducted a phase 1 trial of neoadjuvant nivolumab in patients with resectable non-small cell lung cancer (NSCLC). They analyzed perioperative outcomes to assess the safety of this approach [34]. The study finally included 20 patients. They reported no delays to surgery and described a high conversion rate (54%) from a minimally invasive approach (VATS or RATS) to thoracotomy mainly due to hilar inflammation and fibrosis. However, they concluded that surgery after treatment with nivolumab was not associated with unexpected perioperative morbidity or mortality.

The NADIM trial analyzed the safety, efficacy and feasibility of immunotherapy (nivolumab) combined with neoadjuvant chemotherapy (paclitaxel plus carboplatin) in patients with locally advanced resectable stage IIIA NSCLC (according to the seventh edition of TNM), followed by adjuvant treatment for 1 year with nivolumab. This study included a total of 46 patients and evaluated PFS at 24 months as the primary goal, while the secondary endpoints consisted of evaluating the toxicity profile of the drugs’ combination, the downstaging rate, and the complete resection rate. The PFS was 77.1%, major pathological response (MPR) was 83%, the pathological complete response (pCR) was 63%, and the 1-year OS rate was 97.8%. Treatment-related adverse events during the neoadjuvant treatment occurred in 43 patients (93%), and grade 3 or worse events were found in 14 patients (30%); however, none of these caused surgery delays or deaths [36].

Apart from these objectives, one of the main goals of these studies is to evaluate the safety and feasibility of surgery after this type of treatment.

CheckMate-816 is the first phase 3 study to show a benefit of the neoadjuvant immunotherapy plus chemotherapy combination for resectable NSCLC over standard chemotherapy. Neoadjuvant nivolumab plus chemotherapy resulted in a lower rate of pneumonectomies and showed pCR in 24% of patients compared with 2.2% in patients treated with chemotherapy alone [37,38]. 

Immunotherapy and chemoimmunotherapy, as expected, caused notable alterations in the tumor microenvironment associated with enhanced pathologic responses and survival. 

A neoadjuvant treatment for resectable non-small-cell lung cancer using atezolizumab and carboplatin was demonstrated by Shu et al. in an open-label, multicenter, single-arm, phase 2 trial. They noted that a significant number of patients experienced a large pathological response and that any treatment-related toxicities were tolerable and did not impair surgical resection [39]. 

The main neoadjuvant and adjuvant trials with immunotherapy and/or targeted therapy are shown in Table 1 and Table 2.

## 4. Precision Surgery and Its Application to Thoracic Oncology

### 4.1. The Concept of Precision Surgery

As a direct result of precision medicine, the idea of precision surgery has been introduced in surgical oncology. Precision medicine is a new method for treating and preventing diseases that considers a person’s different genetic makeup, environmental factors, and lifestyle with the final aim of therapy tailored to each individual. With the discovery of targetable molecular alterations and with the concept that cancer is composed of populations of cells with distinct molecular and phenotypic features, this concept has also involved modern oncology and oncological surgery. This branch of surgery is more sophisticated than a mastery of technical maneuvers and involves a deeper understanding of the underlying biological foundation of disease with the final purpose of a targeted, strategic intervention. 

The first reference to precision surgery dates back to 1996, made by Dr. Blake Cady, who summarizes the concept well, stating that the art of surgical oncology is to apply basic principles flexibly to the individual patient [40]. In general, as reported by Lidsky et al., precision surgery aims to apply surgical therapy to those most likely to benefit, and to avoid surgery in those doomed to fail [41]. An example is the work by Passiglia and colleagues that reported, in a meta-analysis, how KRAS and BRAF mutations predict worse recurrence-free and overall survival in patients undergoing resection of colorectal liver metastasis [42]. In order to identify the patients most likely to benefit from surgical treatment, precision surgery suggests considering the KRAS and BRAF mutational status not only as part of the molecular disease characteristics, but mostly in the context of the clinico-pathological disease features.

Concerning the application of precision surgery to thoracic oncology, there are no scientific data or articles reported. Prior to the advent of new molecular therapies, the treatment of NSCLC was well established on the basis of the clinical and/or pathological TNM staging. The advent of effective, targeted therapies for molecularly defined subsets of patients with NSCLC has prompted the need for more extensive genomic characterization, and thoracic oncology and surgery entered an era of therapy co-directed by histology, genotyping, and immunotyping. Recent advancements in the treatment of NSCLC have given patients access to individualized medicines and significant, frequently long-lasting, therapeutic outcomes.

As shown before, the definition of the role of surgery in the context of ICIs or targeted therapies is still to be defined, as large-population trials regarding the application of the new therapies as adjuvant or neoadjuvant treatments are ongoing. 

### 4.2. Targeted Therapies or Immunotherapy in Resectable NSCLC: Which Patients? Adjuvant or Neoadjuvant Setting?

The decision to perform a systemic treatment in an induction or adjuvant setting is still a matter of debate and this concept may also be considered as an item of precision surgery. For example, recent studies have shown that even for early-stage, radically resected NSCLC, micrometastases may be present before surgery and are considered the main factor causing postoperative local or distant recurrence [43]. This is one of the main arguments in favor of also adopting induction treatments in the early stages. In patients with stage IB to stage IIIA NSCLC, a meta-analysis demonstrated an absolute 5% survival improvement at 5 years with preoperative chemotherapy compared to surgery alone [44]. Neoadjuvant therapy has potential advantages: firstly, in vivo assessment of the response to chemotherapy helps identify patients who will potentially benefit from adjuvant treatment; secondly, it provides an early treatment for any potential micrometastatic disease; and finally, it allows a downstaging of the disease with improved resectability. Potential disadvantages include perioperative complications, delay in local treatment secondary to toxicity, and the risk of progression in patients with chemoresistant disease. In particular, the problem of chemoresistance seems to be predictable in the immunotherapy setting. In fact, the first studies demonstrated a correlation between patient prognosis, response rate to treatment and PD-L1 expression [45]. However, these results are still debatable, as other studies have found that patients with high PD-L1 have worse survival rates [46]. Additional research is necessary to justify its use as a prognostic indicator. However, PD-L1 expression may represent a significant factor as a predictive biomarker in the selection of patients for anti-PD-1/PD-L1 treatment. Although some studies found no correlation, many demonstrated greater response rates in patients with high PD-L1 expression in NSCLC tumors compared to low expression [47]. Adjuvant therapy for early-stage NSCLC has the dual goals of treating micrometastatic disease and preventing recurrence. In fact, according to a meta-analysis, adjuvant chemotherapy improved absolute survival by 4% at 5 years for patients with resected early-stage NSCLC compared to surgery alone [48]. However, it is still debatable whether adjuvant treatment is necessary following radical surgical resection of stage IB NSCLC. The ADAURA trial demonstrated that osimertinib, a third-generation EGFR-TKI, regardless of the use of adjuvant standard chemotherapy, significantly improved DFS with tolerable toxicity in patients with fully resected EGFR mutant NSCLC [14]. As a result, osimertinib has been authorized for the adjuvant treatment of patients with resected NSCLC and EGFR mutations. Different clinical trials (e.g., NCT02273375, CheckMate-816, NCT03968419) evaluating the efficacy of adjuvant and neoadjuvant targeted and immunotherapy in early stages are ongoing, the first results will be ready from January 2024.

### 4.3. Redefining the Concept and the Management of Oligometastatic Disease

Another patient population who may particularly benefit from these new therapeutic agents is that of patients affected by advanced stage disease. Indeed, the conventional treatment for advanced NSCLC has been sequential or concurrent chemo-radiotherapy with a dreadful prognosis. However, with molecular therapies, patients with even advanced metastatic NSCLC, for whom surgery was excluded at diagnosis, may have found a window for “curative” or debulking surgery after years of treatment creating a large “grey area” of potentially resectable lung cancers that may benefit from treatment protocols that include surgery. In this setting, the systemic therapy does not have the role of induction treatment, and surgery seems to develop a role of rescue therapy in patients with acquired resistance to targeted drugs. Without any doubt, the inevitable controversies around these complex cases highlight the key role of multidisciplinary tumor boards, preferentially in high-volume centers. In this sense, the exciting results of the new therapies will allow surgeons to play a greater role in more advanced stages with curative intent in mind. Moreover, especially in metastatic disease, genomic profiling seems to play a fundamental role in routine care. Indeed, in addition to the improved survival observed with targeted therapies against EGFR and ALK mutated patients, other studies have demonstrated great results with targeted therapies against BRAF, RET and MET [49]. Israeli et al. reviewed 101 NSCLC patients with negative EGFR/ALK mutations that were tested by NGS. Finally, they discovered clinically relevant genetic changes in 50% of patients, changing the course of treatment for 43 patients. Above all, NGS also identified EGFR mutations in 15 patients with EGFR wild type at conventional testing [50]. 

Regarding metastatic disease, stage IV is highly heterogeneous and, according to current data, survival may vary significantly and is strictly related to the location and quantity of metastases [51].

It is particularly important in the concept of oligometastatic disease; in fact, it has been proposed as an intermediate state between localized and systematically metastasized disease. Clinical investigations conducted in this situation have demonstrated improved survival when regular systemic therapy is combined with radical local therapy (especially surgery) [52]. With this new concept, metastatic NSCLC would no more be considered incurable per definition but must be treated using a multidisciplinary approach in order to clarify not only the localized primary and metastatic tumor lesions, but also the eventual disseminated circulating tumor cells. However, there is currently no clear agreement on the number of metastases and the number of affected organs that may constitute an oligometastatic state. Actually, five or fewer metastases in two or fewer organs have been utilized as a threshold for oligometastases in the majority of reported phase 2–4 clinical trials on the treatment of oligometastatic NSCLC. Moreover, there is still a debate on what is better to treat first and on the timing to perform systemic treatment. As described by Berzenji et al., two different types of approach to oligometastatic disease are possible [53]. The first one includes the initial surgical removal of the primary tumor and, subsequently, the local treatment of metastases (surgery or SBRT) associated with systemic therapy (targeted or immunotherapy are advisable) for the control of micrometastatic disease. 

The second option consists of firstly performing a neoadjuvant treatment followed by a PET-CT scan restaging, and the subsequent resection of the primary tumor and metastatic lesions.

The advantage of upfront surgery is that there is no risk of a decline in the performance status of the patient after an induction treatment and, consequently, a possible delay in surgery. However, with the introduction of the new therapies and their positive response, neoadjuvant treatment may be useful because it could eradicate micrometastasis not detected at the clinical evaluation and can achieve a reduction in tumor volume, promoting less invasive resections. The ETOP-CHESS trial will give us more results in this field. This trial is a single-arm, multicenter phase 2 trial evaluating the efficacy of durvalumab, a platinum-based doublet CT associated with radiotherapy and/or surgery in NSCLC with oligometastatic disease. Durvalumab is used as part of a treatment plan that also includes 4–6 cycles of platinum-based doublet CT and SBRT for all oligometastatic lesions. Radical excision or conclusive radiation therapy (RT) of the initial tumor will be used to complete the treatment for patients whose disease has not advanced at the 3-month FDG-PET/CT restaging. Finally, cases where disease progression is observed in one or a few sites while receiving aggressive systemic therapy should deserve careful attention. In this so-called oligoprogressive disease, evidence on the use of local treatments is scarce despite the rising number of clinical trials on oligometastatic NSCLC. 

Prospective studies evaluating the efficacy of immunotherapy or targeted therapy in an oligometastatic setting are ongoing and will be helpful to understand the correct timing of the different treatments.

### 4.4. Does Neoadjuvant Immunotherapy Complicate Surgical Resection?

Another important concept to keep in mind when facing patients receiving target and/or immunotherapy is the clinical and pathological response of tissues to treatment and its consequences on the surgical approach. 

In fact, following immunotherapy, several authors have reported an intense inflammatory response in the tumor and in the lymph nodes, with the replacement of tumors by fibrotic scar tissue observed upon pathological analysis.

Due to the potential negative effects on surgical viability, the potential use of immunotherapy and targeted therapy as a neoadjuvant treatment has raised a number of concerns. Indeed, inflammatory responses leading to hilar fibrosis could make surgical excision more difficult and technically demanding, thus affecting patients’ morbidity and/or mortality. For example, the IoNESCO trial, a phase 2 study that consisted of the administration of durvalumab as a single agent before surgery, was stopped due to the high rate of severe postoperative complications (tracheal fistula, ARDS, severe pneumonia) [54]. On the contrary, in the retrospective cohort study by Bott and colleagues, all 19 patients who were operated after ICIs treatment had a regular postoperative course, without unexpected morbidity or mortality; however, the surgical procedures were challenging, with more than half of the planned minimally invasive resections converted to open surgery due to hilar inflammation [55]. 

The CheckMate-816 trial reported that the neoadjuvant addition of nivolumab to chemotherapy was tolerable and did not increase post-surgical complications. The surgical outcome analysis demonstrated that the chemoimmunotherapy patients had shorter procedures, needed fewer pneumonectomies, had higher rates of minimally invasive surgery and had fewer conversions to open surgery [37,38]. Caution should also be exerted in case of severe comorbidities that augment the surgical risk, especially in current smokers. In this setting, the correct planning of the surgical procedure is fundamental. The use of minimally invasive approaches, parenchymal sparing techniques, bronchial and vascular reconstruction techniques and the avoidance of pneumonectomies are all well-known principles to improve postoperative outcomes, and may be considered an inherent part of the concept of precision surgery.

Another aspect to consider concerns the increasing evidence that ICIs or targeted therapies administered in the neoadjuvant setting lead to a consistent rate of pathologic complete response after surgery. Therefore, it could be debated whether it is worthwhile to perform surgery with its possible complications on such patients, or if it could be avoided. Probably, at present, the answer remains yes, as we have no preoperative radiological instruments or biomarkers, currently, to confidently identify a patient with no residual disease after treatment. This is certainly an important aspect to investigate, and further studies may help us to answer this question better.

### 4.5. Evaluating Response to Treatment

An additional issue related to the introduction of ICIs or targeted therapies is the definition of response to treatment. Chest-CT scans and PET with FDG are now commonly used to assess tumor response to treatment based on the RECIST criteria. However, in this new contest, even these conventional imaging tools can demonstrate equivocal results. 

In fact, it is debatable whether changes in tumor size, as revealed by radiological images, are indicative of therapeutic response; this is because tumors, in addition to malignant cells, may also contain stroma and inflammatory cells (such as T cells, fibroblasts, macrophages). In particular, peritumoral inflammation is responsible for a phenomenon called pseudoprogression. This is a particular type of clinical response, where the initial growth of the tumor’s size is secondary to the infiltration of inflammatory cells and/or fibrosis due to activation of the immune system. It can appear in patients receiving ICIs or targeted therapies, and it is a significant misleading factor in the evaluation of therapeutic response and efficacy. According to this, Bott et al. firstly described two patients with radiologically stable disease after treatment with two cycles of nivolumab; however, the final pathological exams after surgery reported no residual tumor cells [56].

To overcome these technical difficulties, modified RECIST criteria have been introduced to evaluate the response in patients who receive immunotherapy, the Immune Response Evaluation Criteria in Solid Tumors (iRECIST) introduced in 2017 [57]. The basic principles that define the tumor response evaluation used in RECIST remain the same with iRECIST; however, to define tumor progression a confirmation of tumor enlargement after a minimum of 4 weeks and no later than 8 weeks from the last evaluation is required. Moreover, PET-CT images are fundamental, because by documenting fluorodeoxyglucose uptake (FDG), it is possible to distinguish between pseudoprogression and true progression in some cases [58].

Regarding the pathological evaluation of tumor response, with the advent of new therapies, numerous histologic criteria were reviewed. Firstly the concept of the tumor bed was well-defined, which is the area where the original pre-treatment tumor was located [59].

In this setting, the major three features used for analysis include necrosis, stromal fibrosis and viable tumor. The percent of viable tumors has consistently been shown to be the only prognostically significant histologic indicator.

## 5. Conclusions

Molecular therapies opened a new era in the treatment of NSCLC that is no longer established upon the basis of the clinical and/or pathological TNM staging but is a therapy co-directed by histology, genotyping and immunotyping. This enables patients to receive individualized therapy, resulting in significant and frequently long-lasting treatment outcomes.

In this context, the role of surgery is evolving, moving to a concept of precision thoracic surgery that aims to optimize and individualize patient selection and treatment based on a more sophisticated understanding of cancer, with the goal of giving each patient the most personalized and adequate therapeutic intervention available within our armamentarium.

## Figures and Tables

**Table 1 cancers-15-01571-t001:** Neoadjuvant immunotherapy/target therapy trials.

Trial	Inclusion Criteria	Treatment Arms	Post-Surgery Therapy
EMERGING-CTONG 1103 (NCT01407822)	IIIA-N2 with EGFR-mutation	Erlotinib vs. Cisplatin + Gemcitabine	/
CheckMate 77T (NCT04025879)	II-IIIB (N2)	Standard CT +Nivolumab vs Placebo	Nivolumab vs. Placebo
NADIM study (NCT03081689)	Resectable IIIA NSCLC	Paclitaxel + Carboplatin + Nivolumab vs. Placebo	Nivolumab/ Observation
CheckMate-816 (NCT02998528)	Resectable IB (>4 cm)-IIIA	Nivolumab + Platinum-based CT vs. Platinum-based CT	CT +/− RT
IMpower030 (NCT03456063)	Resectable stage II-IIIB	Atezolizumab + Platinum-based CT vs. Placebo + Platinum based CT	Atezolizumab vs. Placebo
Keynote 671 (NCT03425643)	Stage IIB-IIIA	Pembrolizumab + Platinum-based CT vs. placebo + platinum-based CT	Pembrolizumab vs. Placebo
AEGEAN (NCT03800134)	Resectable stage IIA-IIIB	Durvalumab + Platinum-based CT vs. Placebo + Platinum-based CT	Durvalumab vs. Placebo
NeoADAURA (NCT04351555)	Resectable stage II-IIIB	Osimertinib as single agent or in combination with Platinum-based CT vs.Placebo + Chemotherapy	Osimertinib +/− CT

**Table 2 cancers-15-01571-t002:** Main adjuvant Immunotherapy/targeted therapy trials.

Trial	Inclusion Criteria	Treatment Arms	Primary Endpoints
ADAURA (NCT02511106)	Resected IB-IIIA NSCLC	Osimertinib vs. standard CT	DFS in stage II to IIIA disease
Impower010 (NCT02486718)	IB (4 cm)-IIIA after Adj CT	Atezolizumab	DFS
PEARLS/KEYNOTE-091 (NCT02504372)	Resected stage IB-IIIA	Pembrolizumab	DFS in overall population; DFS in population with PD-L1 > 50%
NADIM-ADJUVANT (NCT04564157)	Resected stage IB-IIIA	CT + Nivolumab, then Nivolumab vs. Platinum-based CT	DFS
ALINA (NCT03456076)	Resected stage IB-IIIA	Alectinib vs. Platinum-based CT	DFS
ALCHEMIST (NCT04267848)	Resected stage II-IIIA	CT + concomitantPembrolizumab, then Pembrolizumab vs. CT + sequentialPembrolizumab, then Pembrolizumab vs. CT	DFS

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
