# Peer review of "Precision Surgery in NSCLC"

_cancers, 2023, doi:10.3390/cancers15051571_

Round 1
Reviewer 1 Report (New Reviewer)
My concerns are appropriately addressed in the manuscript. I have no further comments. I would recommend accepting in current form.
Author Response
We really thank the reviewer for his pertinent comments and the final opinion about our manuscript.
Reviewer 2 Report (Previous Reviewer 1)
The authors addressed the reviewer's concerns.
Author Response
We really thank the reviewer for his pertinent comments and the final opinion about our manuscript.
This manuscript is a resubmission of an earlier submission. The following is a list of the peer review reports and author responses from that submission.
Round 1
Reviewer 1 Report
Suggestions for Authors:
Please correct the required changes in the attached file.
Questions for authors:
Is afatinib reversibly inhibit human epidermal growth factor receptor (EGFR) kinases?
"Afatinib covalently binds to cysteine 797 of the EGFR via a Michael addition. Such covalent binding irreversibly inhibits the tyrosine kinase activity of this receptor." Ref. DOI: 10.1007/s00210-014-0967-3.
Reference 17 stated that "lorlatinibe was associated with more grade 3 or 4 adverse events than crizotinib". However, in the present review, the opposite scenario was described, which should be addressed by the authors for the reviewer's concern.
Reviewer 2 Report
It is too strong to call precission chemotherapy or chemo-imunotherapy combineing with surgery as precision surgery. We all know thouse clinical trials and results combineing surgery and chemo-imunotherapy in adjuvant and neaoadjuvant settings. Discussion was better neoadjuvant or adjuvant treatment before or after surgery is very old and have no right answer. The bulking surgery in oligometastatic disease and useing imuno-chemotherapy in treatment of metastatic disease is controversial with no right answer. Neoadjuvant chemotherapy toxicity in various trials are diferent - from high to moderate as well as number of complications. This is simple overwiev of known clinical trials with recomendation for better patients selection.
Reviewer 3 Report
In this manuscript, the authors discussed the role of surgery in the era of precision oncology. In this review article, the authors summarized genomic profiling in NSCLC and the multimodality approach to NSCLC. I would consider this manuscript for publication after addressing these comments.
1: In lines 35 and 36 authors described the main subtypes of NSCLC, I would recommend adding and including all NSCLC from the WHO lung tumors classification.
2: In line 47, “In fact, thanks……” I will omit this sentence.
3: A liquid biopsy is an emerging and significant role in personalized therapeutic approaches. I would recommend briefly discussing in the introduction. https://www.nature.com/articles/s41416-022-01777-8
4: In Line 89 again thanks to…. I would not use the thanks word.
5: Genomic profiling of lung cancer in metastatic setup is playing an important role in the era of precision oncology, also very important to discuss the role of surgery in metastatic disease.
“The Role of Operation in Metastatic Lung Cancer to Liver and Lungs: A Propensity-Matched National Analysis. Journal of the American College of Surgeons: November 2021 - Volume 233 - Issue 5 - p S261-S262 doi: 10.1016/j.jamcollsurg.2021.07.541”
6: Surgery has the highest impact on early-stage disease and improved clinical outcomes. During COVID-19 pandemic, there was a significant delay in cancer-related surgeries. I would strongly recommend adding a paragraph on delay in surgery and its clinical impact.
https://www.ncbi.nlm.nih.gov/pmc/articles/PMC7711379/
https://jtd.amegroups.com/article/view/66624/html